# SARS-CoV-2 antibody persistence after five and twelve months: A cohort study from South-Eastern Norway

Marjut Sarjomaa[1,2]*, Lien My Diep[3], Chi Zhang[3,4], Yngvar Tveten[5], Harald Reiso[6], Carina Thilesen[7], Svein Arne Nordbø[8,9], Kristine Karlsrud Berg[10], Ingeborg Aaberge[4], Neil Pearce[11], Hege Kersten[12,13], Jan Paul Vandenbroucke[11,14,15], Randi Eikeland[16,17], Anne Kristin Møller Fell[2,17]

1 Department of Infection Control, Telemark Hospital Trust, Skien, Norway, 2 Department of Community Medicine and Global Health and Society, University of Oslo, Oslo, Norway, 3 Oslo Centre for Biostatistics and Epidemiology, Oslo, Norway, 4 Norwegian Institute of Public Health, Oslo, Norway, 5 Department of Clinical Microbiology, Telemark Hospital Trust, Skien, Norway, 6 The Norwegian National Advisory Unit on Tick-borne Diseases, Sørlandet Hospital Trust, Arendal, Norway, 7 Unilabs Laboratory Medicine, Skien, Norway, 8 Department of Medical Microbiology, St. Olavs Hospital, Trondheim University Hospital, Trondheim, Norway, 9 Norwegian University of Science and Technology, Trondheim, Norway, 10 Department of Medical Microbiology, Sørlandet Hospital Trust, Kristiansand, Norway, 11 London School of Hygiene and Tropical Medicine, London, United Kingdom, 12 Department of Research, Telemark Hospital Trust, Skien, Norway, 13 School of Pharmacy, University of Oslo, Oslo, Norway, 14 University of Leiden, Leiden, Netherlands, 15 University of Aarhus, Aarhus, Denmark, 16 Department of Health and Nursing Science, University of Agder, Grimstad, Norway, 17 Department of Occupational and Environmental Medicine, Telemark Hospital Trust, Skien, Norway

* sarm@sthf.no

## Abstract

### Objectives

To assess total antibody levels against Severe Acute Respiratory Syndrome CoronaVirus 2 (SARS CoV-2) spike protein up to 12 months after Coronavirus Disease (COVID-19) infection in non-vaccinated individuals and the possible predictors of antibody persistence.

### Methods

This is the first part of a prospective multi-centre cohort study.

### Participants

The study included SARS-CoV-2 real-time polymerase chain reaction (RT-PCR) positive and negative participants in South-Eastern Norway from February to December 2020. Possible predictors of SARS-CoV-2 total antibody persistence was assessed. The SARS-CoV-2 total antibody levels against spike protein were measured three to five months after PCR in 391 PCR-positive and 703 PCR-negative participants; 212 PCR-positive participants were included in follow-up measurements at 10 to 12 months. The participants completed a questionnaire including information about symptoms, comorbidities, allergies, body mass index (BMI), and hospitalisation.

**Data Availability Statement:** There are legal and ethical restrictions on sharing our dataset. The project is approved by the Regional Committees for Medical and Health Research Ethics (ID 146469),

and by the Data Protection Officers in the participating Hospitals. Our data set is not fully anonymized and has a relative small sample size making identification of individuals possible. The potentially identifying patient information is age, birthdate, location and dates for PCR and antibody tests. However, data requests for the minimal dataset, which includes only the main variables of the final analyses, can be made to the Research Department at The Telemark Hospital Trust, Ulefossvegen 55, 3710 Skien, Norway email: fou@sthf.no.

**Funding:** The authors received no specific funding for this work.

**Competing interests:** The authors have declared that no competing interests exist.

## Primary outcome

The SARS-CoV-2 total antibody levels against spike protein three to five and 10 to 12 months after PCR positive tests.

## Results

SARS-CoV-2 total antibodies against spike protein were present in 366 (94%) non-vaccinated PCR-positive participants after three to five months, compared with nine (1%) PCR-negative participants. After 10 to 12 months, antibodies were present in 204 (96%) non-vaccinated PCR-positive participants. Of the PCR-positive participants, 369 (94%) were not hospitalised. The mean age of the PCR-positive participants was 48 years (SD 15, range 20–85) and 50% of them were male. BMI $\geq$ 25 kg/m$^2$ was positively associated with decreased antibody levels (OR 2.34, 95% CI 1.06 to 5.42). Participants with higher age and self-reported initial fever with chills or sweating were less likely to have decreased antibody levels (age: OR 0.97, 95% CI 0.94 to 0.99; fever: OR 0.33, 95% CI 0.13 to 0.75).

## Conclusion

Our results indicate that the level of SARS-CoV-2 total antibodies against spike protein persists for the vast majority of non-vaccinated PCR-positive persons at least 10 to 12 months after mild COVID-19.

## Introduction

Since the initial outbreak of COVID-19 was reported in Wuhan in December 2019, over 304 million people have been infected worldwide, with over 5.4 million deaths reported by the World Health Organization (WHO) as of 11th January 2022 [1]. The first wave of the pandemic in Norway peaked in March 2020. The second wave started in the autumn of 2020 and the third wave in February 2021. The estimated seroprevalence of SARS CoV-2 antibodies in Norway was 0.6% in the late summer of 2020 and increased to 3.2% in January 2021, after the second wave [2].

The impact of SARS-CoV-2 on human health in individuals and populations depends on multiple factors such as the level of healthcare, diagnostics, therapeutics, social distancing measures such as lockdowns, face masks, working from home, and the availability and coverage of vaccines to control the disease. Understanding the cellular and humoral immunity to COVID-19 is necessary to assess the future course of the pandemic. There is still insufficient data regarding the long-term persistence of antibodies and the level of protective immunity, especially in patients who underwent mild infections and those who were not hospitalised [3–9]. Hence, there is a need for studies on antibody kinetics to improve our understanding of humoral immunity following COVID-19 infections.

The gold standard for antibody test assays has not been determined, and numerous immunoassays have been developed [10]. IgG antibodies against SARS-CoV-2 are highly sensitive markers 7–14 days after symptom onset [8, 11]. Antibodies are detected in 90% of individuals after two weeks and are highly correlated with neutralising antibodies [8, 12]. High sensitivity and specificity are important for all serological assays, and the specificity of an antibody test might be an issue when the infection prevalence is low [13]. Some smaller studies early in the

pandemic showed that antibodies declined within a few months after infection [14–16]. Antibody levels against the nucleocapsid protein have been shown to decline more rapidly than antibodies against the spike protein [12]. Furthermore, the kinetics and protective immunity between anti-nucleocapsid and anti-spike antibodies may differ [17]. Data from large, systematic, and quantitative follow-up studies of antibodies for longer than six to eight months are limited [18, 19].

The study of antibody persistence is still important, despite vaccination, as antibody longevity during an ongoing pandemic is of scientific interest, as well as being particularly relevant for providing correct and informed vaccination strategies to those who have had an infection. However, there is still a lack of knowledge regarding the longevity of protective immunity.

Our aim was to assess the SARS-CoV-2 total antibody levels against spike protein up to 12 months after COVID-19 infection in non-vaccinated individuals, and identify possible predictors for antibody persistence [20].

## Materials and methods

### Study design and setting

This study is the first part of the COVITA -project (COVID-19 Telemark and Agder study) which is a prospective multi-centre cohort study. We here present population characteristics and antibody persistence of PCR -positive (PCR+) and PCR -negative (PCR-) participants. Further, results from the first follow-up of the PCR+ participants regarding possible predictors for antibody change are shown. Adults aged 18 years or older residing in South-Eastern Norway (Agder and Telemark counties) were considered eligible for inclusion in the study. Participants were recruited from all hospitals in the region, municipality laboratories, and test centres between February 28 and December 17, 2020. The study included SARS-CoV-2 RT-PCR -positive and -negative participants regardless of symptoms. We included participants who performed a PCR test in the inclusion period, and used the first PCR test result. For each PCR+ participant, we aimed to select two PCR- participants matched by residency and time for the PCR test to reduce the probability of health care-seeking bias. In this study, we included PCR- participants to allow comparison of population characteristics between PCR+ and PCR—participants, and to assess presence of antibodies also among PCR- participants.

Participants who did not consent or were unable to answer a questionnaire (Norwegian language) were excluded. COVID-19 vaccination was introduced in January 2021, and post-vaccine samples were excluded in this study.

The official Norwegian testing criteria for SARS-CoV-2 changed over time but were the same for the PCR+ and PCR- participants. In the first wave of the pandemic, PCR testing was restricted to patients with symptoms. In the second wave, PCR testing was also applied to close contacts and asymptomatic persons during outbreaks. All participants were invited to the sampling of antibodies and were asked to fill in the self-reporting questionnaire simultaneously, three to five months (time point 1 = T1) after the PCR test. The group of non-vaccinated PCR+ participants were invited for follow-up 10 to 12 months (time point 2 = T2) after the PCR test.

Participants gave their informed, written consent, and no financial incentives were offered.

### Data sources/Measurement

**Questionnaire.** Questions from the Norwegian Health Institute COVID-19 questionnaire and the validated Telemark-study questionnaire were used [21–24], in addition to questions provided by the study group. A user representative was involved in the process of piloting the questionnaire. Questions are shown in S1 Table.

The questionnaire consisted of questions about demographic data, income, education, smoking habits, hospitalisation, and comorbidities such as asthma and chronic obstructive pulmonary disease (COPD), other lung diseases, cancer, heart disease, diabetes, hypertension, musculoskeletal disease, any other disease and pollen allergy. Questions about symptoms included the presence or absence of cough, runny nose, stuffy nose, sore throat, dyspnoe, headache, fever with chills or sweating, abdominal pain, nausea, diarrhoea, impaired sense of smell and taste, myalgia, and dizziness. Questions regarding fatigue and reinfection were also included.

Self-reported vaccination data were available and checked against data from the National Immunisation Register (SYSVAK). Additional demographic data (age, sex, and time and place for SARS-CoV-2 PCR) were also registered.

**Laboratory methods.** Venous blood samples were obtained at recruitment, three to five months after the PCR test (T1), and again after 10–12 months (T2). All serum samples were prepared from whole blood following centrifugation for 10 min at a minimum of 1800 g at room temperature and stored at -80 $^{0}$C until further analysis. Total immunoglobulin levels were analysed at Telemark Hospital using the Siemens Advia Centaur XP SARS-CoV-2 Total assay for the qualitative detection of total antibodies (IgM and IgG) in human serum. On a large panel of blood samples, the Siemens assay achieved a sensitivity and specificity of at least 98% [10]. The assay is a fully automated 1-step antigen sandwich immunoassay using acridinium ester chemiluminescent technology and recombinant SARS-CoV-2 S 1 receptor binding domain (RBD) antigen. The total antibody ranged from 0 to 9.99, and $\geq$ 10 was the upper limit of the assay. The threshold for reactivity was $\geq$ 1.0 Index. To make it possible to interpret changes in antibody levels, antibody levels were categorised into four categories: negative (0–0.79), low (0.80–1.99), intermediate (2.00–9.99), and high ($\geq$10).

To reduce the possibility of false positive results, low antibody levels between 0.80 and 1.99 Index value were reanalyzed at St. Olavs Hospital in Trondheim using the Elecsys Cobas SARS-CoV-2 total antibody test (Roche) and BioPlex 2200 SARS-CoV-2 IgG Panel (Bio-Rad).

**Ethics approval and consent to participate.** All participants provided written informed consent before inclusion. The Study was approved by the Regional Committee for Medical and Health Research Ethics of South East Norway A (ID 146469), the Norwegian Centre for Research Data (NSD), and the Data Protection Officers in the participating hospitals.

**Statistical analysis.** Mean, standard deviation (SD), medians, interquartile ranges (IQR), and minimum and maximum values were reported for continuous data as appropriate. Categorical data were reported as frequencies and percentages.

The difference between the observed antibody values at T1 and T2 were calculated. The changes that were greater than the maximum level of 10 Index could not be quantified due to the limitations of the assay. Hence, for further analysis, the participants were categorised into four antibody groups: decreased, unchanged, increased, and maximum level $\geq$ 10. An increase or decrease of at least 20% in antibody values from T1 to T2 was defined as a significant change; otherwise, the participant was assigned to the unchanged antibody group. Chi-squared and Fisher's exact tests were used to examine differences in proportions between the decreased antibody group and the other groups (unchanged, increased, and maximum antibody level $\geq$ 10) for binary variables. For continuous and ordinal variables, the Kruskal-Wallis test and the One-Way Anova was used to study the difference. Total symptom and comorbidity scores were calculated by adding the number of symptoms or comorbidities for each participant, and used instead of single variables in the multivariate analysis.

Logistic regression models were used to study the possible predictors for changes in antibody levels. Univariate analysis was performed for each predictor, adjusting for age and sex. Then, a multivariate analysis including all predictors was performed, and the standard error of

the effect estimate was compared to that of the corresponding univariate analysis for each predictor to check for collinearity. Odds ratio (OR) was reported with 95% confidence intervals (CI) for association between decreased antibody level and predictors. The analyses were performed with IBM SPSS 27 for Windows (IBM Corp. Released 2020. IBM SPSS Statistics for Windows version 27.0. Armonk, NY: IBM Corp), Stata/SE 16.1 for Windows and R (version 4.2).

## Results

Between February 28 and December 17, 2020, 656 PCR+ participants and 923 PCR- controls matched in time and location to the PCR+ participants were eligible. In summary, 232 PCR+ participants and 92 PCR-participants did not meet the inclusion criteria. Nine PCR+ and 16 PCR- participants were vaccinated prior to the first antibody sampling and were excluded from the study. The number of individuals assessed for eligibility and individuals included in the study is shown in Fig 1.

The demographics of the study population by baseline are presented in **Table 1**.

Except for asthma (18%), the prevalence of self-reported comorbidities was low for all participants. However, PCR+ participants reported more symptoms such as dyspnoea, headache, fever with chills or sweating, abdominal pain, nausea or diarrhoea, impaired sense of taste and smell, myalgia, and dizziness compared to the PCR- participants. In contrast, PCR- participants reported more nasal symptoms (runny/stuffy nose), sore throat, and pain upon swallowing compared to the PCR+ participants. Coughing was similar in both groups. Self-reported moderate or serious fatigue was more frequent among PCR+ than PCR- participants. Only 22 (6%) of the PCR+ participants were hospitalised due to COVID-19. There were no self-reported reinfections during the study period.

The duration and changes in total antibody levels against the SARS-CoV-2 spike protein are shown in **Table 2**.

Among the PCR+ participants, 366 (94%) had total levels of antibodies $\geq$ 0.8, and 208 (53%) had total levels of antibodies at the upper limit of quantification three to five months after the positive PCR test. Twenty participants with uncertain antibody values were reanalysed; three of them showed no antibodies in the secondary test and were defined as false positives in the first test. Among the 212 participants with antibody measurements at both T1 and T2, 204 (96%) had antibodies $\geq$ 0.8, and 113 (53%) had values above the upper limit of quantification of the total antibody at both measurements. The time between PCR and antibody measurements is shown in S2 Table.

The distribution of variables in PCR+ participants with decreased, unchanged, increased and antibody levels $\geq$ 10 at time points T1 and T2 are shown in **Table 3**.

Univariate and multivariate regression analyses were performed to assess the possible predictors for antibody levels after 10–12 months, and to compare the group with decreased antibody levels with the other groups (increased, unchanged, and antibody $\geq$ 10), as shown in **Table 4**.

## Discussion

Total antibodies (Ig M and Ig G) levels against the SARS-CoV-2 spike protein were detectable in 96% of the participants up to 12 months after a positive PCR test. BMI $\geq$ 25 kg/m$^2$ was positively associated with decreased antibody level (OR 2.34, 95% CI 1.06 to 5.42). Participants with higher age and self-reported initial fever with chills or sweating were less likely to have decreased antibody levels (age: OR 0.97, 95% CI 0.94 to 0.99; fever: OR 0.33, 95% CI 0.13 to 0.75). There was no association between antibody persistence and gender. Past smoking

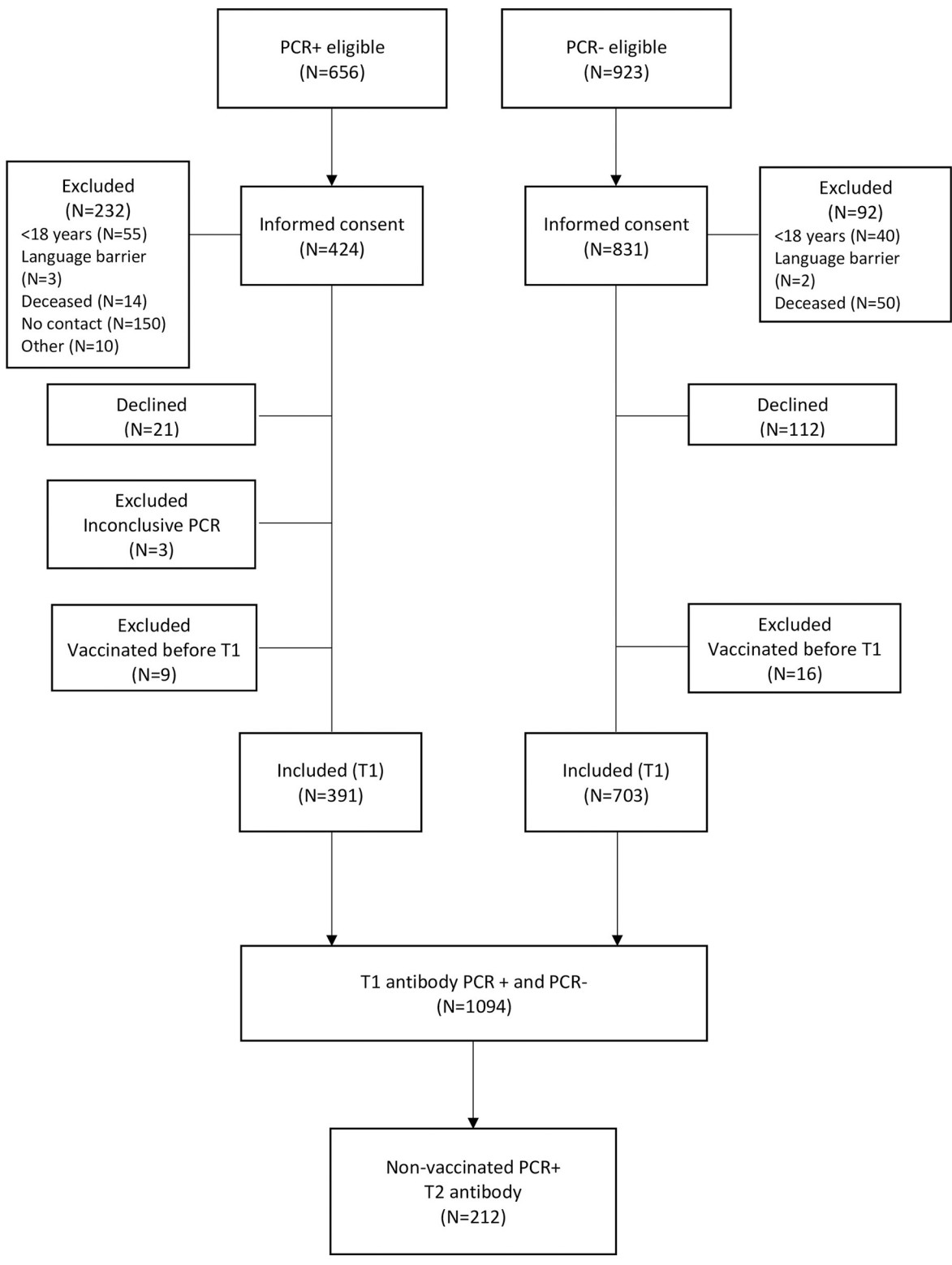

**Fig 1. Flow chart for the study.**

**Table 1. Characteristics of the PCR positive (PCR+) and PCR negative (PCR-) participants at the first antibody test (T1).**

| Characteristics | Total (N = 1094) | PCR+ (N = 391) | PCR-(N = 703) |
|---|---|---|---|
| | N (%) | N (%) | N (%) |
| **Age in years, mean (SD)** | 47.5 (14.6) | 47.9 (15.0) | 47.3 (14.3) |
| N (range) | 1094 (19–98) | 391 (20–85) | 703 (19–98) |
| **BMI in kg/m², median (IQR)** | 25.6 (23.2–29.1) | 25.6 (23.2–28.7) | 25.6 (23.3–29.3) |
| N (range) | 1057 (15.7–65.6) | 374 (15.7–42.8) | 683 (15.7–65.5) |
| BMI in kg/m² ≥ 25 | 598 (54.7) | 206 (55.1) | 392 (57.4) |
| **Gender, males** | 435 (39.8) | 196 (50.1) | 239 (34.0) |
| **Income** | | | |
| <500 000 NOK[a] | 216 (19.7) | 68 (17.4) | 148 (21.1) |
| 500 000–1 000 000 NOK | 474 (43.3) | 162 (41.4) | 312 (44.4) |
| ≥ 1 000 000 NOK | 365 (33.4) | 137 (35.0) | 228 (32.4) |
| **Education** | | | |
| Primary & secondary school | 116 (10.6) | 40 (10.2) | 76 (10.8) |
| High school & certificate | 362 (33.1) | 146 (37.3) | 216 (30.7) |
| University less than 4 years | 342 (31.3) | 128 (32.7) | 214 (30.4) |
| University > 4 years | 255 (23.3) | 66 (16.9) | 189 (26.9) |
| **Smoking** | | | |
| Non-smoker | 527 (56.0) | 213 (61.5) | 314 (52.8) |
| Past smoker | 302 (32.1) | 103 (29.8) | 199 (33.4) |
| Occasional and daily smoker | 112 (11.9) | 30 (8.7) | 82 (13.8) |
| **Comorbidity** | | | |
| Asthma | 195 (17.8) | 61 (15.6) | 134 (19.1) |
| COPD | 34 (3.1) | 7 (1.8) | 27 (3.8) |
| Other chronic lung disease | 44 (4.0) | 16 (4.1) | 28 (4.0) |
| Cancer | 37 (3.4) | 7 (1.8) | 30 (4.3) |
| Heart disease | 48 (4.4) | 16 (4.1) | 32 (4.6) |
| Diabetes | 47 (4.3) | 20 (5.1) | 27 (3.8) |
| Hypertension | 110 (10.1) | 39 (10.0) | 71 (10.1) |
| Musculoskeletal disease | 63 (5.8) | 17 (4.3) | 46 (6.5) |
| Any other disease | 194 (17.7) | 55 (14.1) | 139 (19.8) |
| No disease | 554 (50.6) | 211 (54.0) | 343 (48.8) |
| Pollen allergy | 298 (27.2) | 96 (24.6) | 202 (28.7) |
| **Symptoms** | | | |
| Cough | 511 (46.7) | 181 (46.3) | 330 (46.9) |
| Running nose | 470 (43) | 124 (31.7) | 346 (49.2) |
| Stuffy nose | 416 (38.0) | 123 (31.5) | 293 (41.7) |
| Sore throat | 531 (48.5) | 161 (41.2) | 370 (52.6) |
| Pain in swallowing | 239 (21.8) | 53 (13.6) | 186 (26.5) |
| Dyspnoea | 417 (38.1) | 296 (50.1) | 221 (31.4) |
| Headache | 565 (51.6) | 250 (63.9) | 315 (44.8) |
| Fever | 503 (46.0) | 254 (65.0) | 249 (35.4) |
| Fever with chills or sweating | 311 (28.4) | 163 (41.7) | 148 (21.1) |
| Abdominal pain, nausea or diarrhoea | 250 (22.9) | 117 (29.9) | 133 (18.9) |
| Impaired sense of smell or taste | 360 (32.9) | 260 (66.5) | 100 (14.2) |
| Myalgia | 459 (42.0) | 227 (58.1) | 232 (33.0) |
| Dizziness | 318 (29.1) | 167 (42.7) | 151 (21.5) |

(*Continued*)

**Table 1.** (Continued)

| Characteristics | Total (N = 1094) | PCR+ (N = 391) | PCR-(N = 703) |
|---|---|---|---|
| | N (%) | N (%) | N (%) |
| **Fatigue, moderate and serious** [b] | 132 (12.1) | 104 (26.6) | 28 (4.0) |

[a] One NOK = 0.096 Euro

[b] Fatigue categories:1 = no, 2 = light, 3 = moderate, 4 = serious

SD: Standard deviation

IQR: Interquartile range

history was negatively associated with decreased antibody levels (OR 0.37, 95% CI 0.12 to 0.96) in our study, but this finding should be interpreted with caution because there were few smokers.

Among the PCR negative participants, nine had total antibodies ≥ 0.8 at T1, and after initial symptoms, they did not report reinfection between the time of the PCR test and the antibody measurement. The most likely explanation is that they had COVID-19 and a false-negative PCR test, a test taken incorrectly or a low virus level at the time of the PCR test.

We did not detect any self-reported reinfections during the study period. A recent large prospective multicentre cohort study from all the regions in England showed an 84% lower risk of reinfection after seven months of follow-up post SARS-CoV-2 infection among health-care workers (HCWs) with positive antibodies tested with a range of assays [25]. The correlation between antibody levels and protective immunity remains unclear.

The results for antibody persistence after 10 to 12 months in our study were in accordance with those of several studies with a shorter follow-up time [9, 19, 21]. A seroprevalence study from Iceland reported that pan-IgG antibodies did not decline within four months after the diagnosis of SARS- CoV-2 infection in (91% of the included subjects [26]. Furthermore, a study from the USA initially assessing 188 SARS-CoV-2 PCR-positive subjects showed durable spike IgG titres at six to eight months after symptom onset for 36 out of 40 individuals at follow-up [19]. Recently, a peer-reviewed study by Turner et al. showed that in a follow-up of 42

**Table 2. The duration and changes in total antibody levels against the SARS-CoV-2 spike protein measured with Index value three to five months (T1) and 10 to 12 months (T2) after PCR test for PCR positive (PCR+) and PCR negative (PCR-) participants.**

| Time point to antibody measurement | Antibody level category in Index value | Total | PCR+ | PCR- |
|---|---|---|---|---|
| | | N (%) | N (%) | N (%) |
| **T1 antibody** | 0.00–0.79 | 718 (65.0) | 25 (6.4) | 693 (98.7) |
| **T1 antibody** | 0.80–1.99 | 27 (2.5) | 26 (6.6) | 1 (0.1) |
| **T1 antibody** | 2.00–9.99 | 135 (12.3) | 132 (33.8) | 3 (0.4) |
| **T1 antibody** | ≥10 | 213 (19.5) | 208 (53.2) | 5 (0.7) |
| **T2 antibody** | 0.00–0.79 | | 8 (3.8) | |
| **T2 antibody** | 0.80–1.99 | | 14 (6.6) | |
| **T2 antibody** | 2.00–9.99 | | 63 (29.7) | |
| **T2 antibody** | ≥10 | | 127 (59.9) | |
| **Decreased antibody level, T1 to T2** | | | 39 (18.4) | |
| **Unchanged antibody level, T1 to T2** | | | 35 (16.5) | 2[a] |
| **Increased antibody level, T1 to T2** | | | 25 (11.8) | 1[a] |
| **Antibody level ≥ 10 at T1 & T2** | | | 113 (53.3) | |

[a] Percent is not reported due to low number of participants with antibodies in this group.

**Table 3. The distribution of variables in PCR+ participants with detectable antibodies grouped as decreased, unchanged, increased, and antibody level $\geq$ 10 at time point T1 and T2.**

| Variable of interest[a] | Decreased N = 39 | Unchanged and increased N = 62 | Antibody $\geq$ 10 N = 114 | p-value[b] |
|---|---|---|---|---|
| | N (%) | N (%) | N (%) | |
| **Age in years**, mean (SD) | 44.5 (12.7) | 47.5 (13.8) | 52.6 (14.6) | 0.003 |
| N (range) | 39 (22–78) | 62 (21–81) | 114 (21–85) | |
| **BMI $\geq$ 25.0 kg/m$^2$** | 26 (68.4) | 24 (40.0) | 65 (59.1) | 0.011 |
| **Gender**, males | 20 (51.3) | 29 (46.8) | 65 (57.0) | 0.417 |
| **Income** | | | | |
| <500 000 NOK[c] | 8 (21.6) | 7 (11.5) | 13 (11.8) | 0.605 |
| 500 000–1 000 000 NOK | 11 (29.7) | 22 (36.1) | 49 (44.5) | |
| $\geq$ 1 000 000 NOK | 18 (48.6) | 32 (52.5) | 48 (43.6) | |
| **Education** | | | | |
| Primary, secondary, and high school | 17 (44.7) | 23 (37.7) | 51 (45.1) | 0.588 |
| University less than 4 years | 14 (36.8) | 24 (39.3) | 41 (36.3) | |
| University more than 4 years | 7 (18.4) | 14 (23.0) | 21 (18.6) | |
| **Smoking** | | | | |
| Non-smoker | 28 (80.0) | 36 (64.3) | 57 (53.8) | 0.080 |
| Past smoker | 5 (14.3) | 15 (26.8) | 39 (36.8) | |
| Occasional and daily smoker | 2 (5.7) | 5 (8.9) | 10 (9.4) | |
| **Comorbidity** | | | | |
| Asthma | 5 (13.5) | 9 (14.8) | 20 (17.9) | 0.772 |
| COPD | 0 | 1 (1.6) | 4 (3.7) | 0.583 |
| Other chronic lung disease | 1 (2.6) | 2 (3.3) | 4 (3.5) | 1.000 |
| Cancer | 1 (2.6) | 1 (1.6) | 1 (0.9) | 0.753 |
| Heart disease | 2 (5.3) | 3 (4.9) | 7 (6.2) | 1.000 |
| Diabetes | 0 | 1 (1.6) | 10 (8.9) | 0.040 |
| Hypertension | 2 (5.3) | 2 (3.3) | 16 (14.2) | 0.048 |
| Musculoskeletal disease | 2 (5.3) | 2 (3.3) | 4 (3.5) | 0.790 |
| Any other disease | 7 (18.4) | 6 (9.8) | 7 (18.4) | 0.471 |
| **Comorbidity sum**, mean (SD) | 0.5 (0.7) | 0.4 (0.7) | 0.7 (0.9) | 0.100 |
| **Pollen allergy** | 6 (16.2) | 15 (24.6) | 30 (26.8) | 0.504 |
| **Symptoms by the time of diagnosis of covid-19** | | | | |
| Cough | 15 (42.9) | 26 (47.3) | 62 (61.4) | 0.083 |
| Running nose | 13 (37.1) | 17 (30.4) | 29 (28.7) | 0.646 |
| Stuffy nose | 15 (42.9) | 18 (32.1) | 30 (28.8) | 0.309 |
| Sore throat | 14 (41.2) | 22 (39.3) | 48 (47.1) | 0.607 |
| Pain upon swallowing | 4 (11.4) | 7 (12.5) | 15 (14.6) | 0.921 |
| Dyspnoea | 21 (60.0) | 30 (53.6) | 66 (64.7) | 0.390 |
| Headache | 28 (82.4) | 43 (76.8) | 70 (68.0) | 0.197 |
| Fever | 24 (68.6) | 42 (75.0) | 82 (79.6) | 0.400 |
| Fever with chills or sweating | 10 (25.6) | 24 (38.7) | 58 (50.9) | 0.017 |
| Abdominal pain, nausea or diarrhoea | 14 (40.0) | 13 (23.6) | 34 (33.7) | 0.231 |
| Impaired sense of smell or taste | 29 (82.9) | 41 (73.2) | 80 (77.7) | 0.561 |
| Myalgia | 21 (60.0) | 29 (52.7) | 73 (70.9) | 0.068 |
| Dizziness | 14 (40.0) | 25 (45.5) | 56 (54.9) | 0.245 |
| **Number of symptoms**, mean (SD) | 6.3 (2.7) | 6.0 (2.2) | 6.8 (2.5) | 0.177 |

*(Continued)*

**Table 3.** (Continued)

| Variable of interest[a] | Decreased N = 39 | Unchanged and increased N = 62 | Antibody ≥ 10 N = 114 | p-value[b] |
|---|---|---|---|---|
| | N (%) | N (%) | N (%) | |
| **Fatigue** [e], moderate and serious | 7 (20.0) | 10 (17.0) | 42 (38.5) | 0.006 |

[a] Table 3 includes 212 PCR+ participants and three PCR- participants with detectable antibodies.

[b] P-value for comparison between decreased and the group (unchanged, increased, and antibody ≥ 10).

[c] NOK = Norwegian kroner (One NOK = 0.096 Euro)

[d] COPD = Chronic obstructive pulmonary disease

[e] Fatigue categories:1 = no, 2 = light, 3 = moderate, 4 = serious.

COVID-19 convalescent patients, spike IgG antibodies were detectable at least 11 months after infection [9]. A recent cross-sectional study from USA showed antibodies at median 8.7 months after COVID-19 diagnosis [27].

Other studies that used different assays have shown contradictory results for antibody persistence with rapidly declining antibodies [15, 22]. There is a large heterogeneity in the test

**Table 4. Possible predictors for decreased antibodies after 10 to 12 months compared with increased, unchanged and antibody ≥ 10 using univariate analysis adjusted for age and sex, and multivariate analysis adjusted for all variables.**

| | | Univariate analysis | | | | Multivariate analysis | | | |
|---|---|---|---|---|---|---|---|---|---|
| | | OR | 2.5% | 97.5% | p-value | OR | 2.5% | 97.5% | p-value |
| **Age (numeric)** | | 0.97 | 0.94 | 0.99 | 0.01 | 0.97 | 0.93 | 1.00 | 0.030 |
| **Gender** | Female | ref | | | | ref | | | |
| | Male | 0.92 | 0.46 | 1.85 | 0.81 | 0.77 | 0.32 | 1.85 | 0.563 |
| **BMI kg/m²** | < 25 | ref | | | | ref | | | |
| | ≥ 25 | 2.34 | 1.06 | 5.42 | 0.04 | 2.15 | 0.9 | 5.41 | 0.093 |
| **Education** | Below University | ref | | | | ref | | | |
| | University < 4 years | 0.79 | 0.35 | 1.76 | 0.57 | 0.60 | 0.23 | 1.50 | 0.278 |
| | University ≥ 4 years | 0.83 | 0.30 | 2.15 | 0.71 | 0.50 | 0.14 | 1.62 | 0.265 |
| **Income Norwegian kroner** | < 500 000 | ref | | | | ref | | | |
| | 500 000–1000 000 | 0.47 | 0.16 | 1.41 | 0.17 | 0.61 | 0.19 | 2.12 | 0.425 |
| | > 1000 000 | 0.62 | 0.23 | 1.77 | 0.35 | 0.82 | 0.25 | 2.87 | 0.743 |
| **Smoking** | Non-smoker | ref | | | | ref | | | |
| | Past smoker | 0.37 | 0.12 | 0.96 | 0.06 | 0.34 | 0.11 | 0.93 | 0.049 |
| | Daily and occasional smoker | 0.37 | 0.06 | 1.46 | 0.21 | 0.37 | 0.05 | 1.62 | 0.240 |
| **Symptoms** | | | | | | | | | |
| **Symptoms score[a] (numeric)** | | 1.0 | 0.85 | 1.08 | 0.48 | 0.97 | 0.84 | 1.11 | 0.612 |
| Dyspnoea | No | ref | | | | | | | |
| | Yes | 1.02 | 0.50 | 2.08 | 0.97 | not included | | | |
| Fever with chills/sweating | No | ref | | | | | | | |
| | Yes | 0.33 | 0.13 | 0.75 | 0.01 | not included | | | |
| Abdominal pain/diarrhoea/ nausea | No | ref | | | | | | | |
| | Yes | 1.69 | 0.78 | 3.59 | 0.18 | not included | | | |
| Loss of smell and taste | No | ref | | | | | | | |
| | Yes | 1.17 | 0.53 | 2.74 | 0.71 | not included | | | |
| **Comorbidity score[a]** | | 1.02 | 0.61 | 1.62 | 0.93 | 1.08 | 0.61 | 1.84 | 0.774 |

[a] Total symptom and comorbidity scores were calculated by adding the number of symptoms or comorbidities for each participant, and used instead of single variables in the multivariate analysis.

performance of different immunoassays [28]. Some studies have investigated nucleocapsid antibodies, which have been shown to decline faster than spike antibodies [29, 30]. A study of 97 participants in Korea showed declining nucleocapsid IgG antibody levels with reduction rates of 46% from six weeks to six months after diagnosis [29].

Lower persistence of IgG antibodies has been reported in asymptomatic compared to symptomatic COVID-19 patients two to three months after the PCR-confirmed SARS-CoV-2 infection in a study of 37 participants from China [15]. Accordingly, a study from Italy showed a declining antibody response with three different serological assays in 20 individuals with mild symptoms two months after the initial symptom onset [28].

Many of the published studies on antibody levels have focused on specific populations such as HCWs or hospitalised patients, and these results may not be representative of the general population [18, 31]. In a Swedish longitudinal seroprevalence study among 355 HCWs, 98% had antibodies against spike protein at least four months post infection [30]. Antibodies were lower among HCWs than among 59 hospitalized COVID-19 patients. Most HCWs had mild symptoms prior to inclusion in the study.

Previous studies reporting factors associated with longevity of antibodies show a decline in antibodies or seronegativity among asymptomatic patients or patients with mild symptoms. Van Elslande et al. showed in their study that SARS-CoV-2 anti-nucleocapsid antibody levels were significantly lower in those with asymptomatic and mild SARS-CoV-2 infection than in those with severe disease, and that 61% of them became seronegative within 6 months post-PCR [17]. Hence, our findings of up to 12 month persistence in 96% of mainly non-hospitalised persons with mild symptoms are interesting. However the difference may be explained by the use of antibody against spike protein instead of nucleocapsid.

Our study agrees with other studies concerning the association between age and antibody persistence. In a 180 days follow-up study of 164 infected patients, Chia et al. found that antibody persistence was associated with disease severity, older age, and more comorbidities compared with those who had rapidly declining antibodies, although a greater proportion (11 of 19) of asymptomatic individuals were in the seronegative group [32].

BMI $\geq$ 25 kg/m$^2$ was strongly associated with decreased antibody level in our study. This was not in agreement with a recent Norwegian study that showed the severity of initial illness, older age, and higher BMI were independently associated with increased antibody levels two months after infection [33]. A large prospective cohort study from the United States showed homogenous immune activity across BMI categories [34]. A large retrospective study from Israel showed that the peak level of neutralising antibodies was associated with obesity, with the highest level in patients who were severely obese [35]. The study did not include any follow-up antibodies. A study from Turkey showed no effect of BMI on antibody titres, but the sample size was small and the follow-up period was only 60 days [36]. Nevertheless, obesity was a risk factor for COVID-19 infection severity in many studies as shown in a meta-analysis of 42 studies [37]. It is possible that the diversity of the results of the association between BMI and antibody persistence after COVID-19 infection is dependent on disease severity.

There was no association between gender, income, education, and antibody persistence in our study. Socioeconomic differences in Norway are relatively small, and most of the included participants were from the middle class and above, as reflected in the high levels of education and income among the participants.

Our study has several strengths and limitations. A strength of our study is the relatively large and unselected study cohort recruited during the first and second pandemic waves in Norway. The time for the antibody test was related to the PCR test, and not the onset of symptoms. Another strength is the longitudinal design with a relatively long follow-up time.

Validated serology assays were used, and the participants answered the questionnaire on the same day as the serum sampling.

A limitation of our study is that changes in antibody levels above the assay's upper limit value 10 Index could not be determined, but it is probably in the group with low antibody levels that significant changes will appear. Direct comparison of the current study's results with those of other studies is difficult because assays targeting different antigens are used in the available studies [10]. Further, differences in study populations, severity of illness, and frequency of antibody sampling make comparisons across studies challenging [31]. In our study, the same assay was used for all participants and as a quality control, all PCR- participants with detectable antibodies and all PCR+ participants with low levels of antibodies were retested with a second assay. Using two different assays was considered useful in low seroprevalence countries like Norway because of the possibility of false-positive tests [14].

The use of a self-reported questionnaire may be considered a limitation of the study because of possible recall bias. Further, the participants might not understand the questions as intended, and the response rates differed for the questions used. However, all questions included in the questionnaire were chosen from validated or frequently used questionnaires [21–24, 38].

Measurements of cellular immunity and neutralising antibodies were not feasible in this study, hence no firm conclusions can be drawn regarding the immune status post COVID-19 [19]. T-cell responses in SARS-CoV-2 patients have been detected in recovered patients without an antibody response [39]. Early T-cell response may have an important role in protection against reinfections and recovering from SARS-COV-2 infection [40]. T-cell immunity is robust, and its memory might be longer-lasting [41].

## Conclusion

In this Norwegian study of mainly non-hospitalized SARS CoV-2 PCR+ participants, 96% had high antibody levels one year after the PCR test. No reinfections were detected in this period. BMI $\geq$ 25 kg/m$^2$ was positively associated with decreased antibody levels (OR 2.34, 95% CI 1.06 to 5.42). Participants with higher age and self-reported initial fever with chills or sweating were less likely to have decreased antibody levels (age: OR 0.97, 95% CI 0.94 to 0.99; fever: OR 0.33, 95% CI 0.13 to 0.75). We did not detect an association between antibody persistence and gender. Our results indicate that the level of total antibodies against spike protein following mild COVID-19 persists for at least 10 to 12 months.

## Supporting information

**S1 Table. Questions from the questionnaire concerning demographic data, hospitalisation, comorbidities, symptoms and fatigue.**
(DOCX)

**S2 Table. The median time between positive PCR and antibody measurements.** The median time between positive PCR and T1 antibody measurement was 127 days (91–153) and between positive PCR and T2 antibody 310 days (291–329). Among the PCR negative participants, the median time between PCR test and T1 total antibody was 200 days (114–217).
(DOCX)

## Acknowledgments

We thank Sølvi Noraas (Department of Clinical Microbiology, Kristiansand, Norway) for organising of the serological analysis in Agder. The authors are deeply grateful to Trude

Belseth Sanden, Astrid Bjørkeid, June Bakstevold, Gølin Finkenhagen Gundersen, Emile van Gelderen, Elin Skjørvold Christensen, Louise Myrland, Signe Seljåsen, Mona Brekke, Siv Stigen, Anne Cecilie Tveiten, Oda Eikeland Myrnes and Siri Cathrine Rølland for their essential assistance for data collection and analysis. The authors would like to express their gratitude to all the participants in this study and to the patient representatives.

## Author Contributions

**Conceptualization:** Marjut Sarjomaa, Yngvar Tveten, Carina Thilesen, Neil Pearce, Hege Kersten, Jan Paul Vandenbroucke, Randi Eikeland, Anne Kristin Møller Fell.

**Data curation:** Marjut Sarjomaa, Harald Reiso, Kristine Karlsrud Berg, Anne Kristin Møller Fell.

**Formal analysis:** Lien My Diep, Chi Zhang.

**Funding acquisition:** Anne Kristin Møller Fell.

**Investigation:** Marjut Sarjomaa, Yngvar Tveten, Harald Reiso, Carina Thilesen, Kristine Karlsrud Berg, Randi Eikeland.

**Methodology:** Neil Pearce, Jan Paul Vandenbroucke, Anne Kristin Møller Fell.

**Project administration:** Anne Kristin Møller Fell.

**Resources:** Marjut Sarjomaa, Yngvar Tveten, Carina Thilesen, Svein Arne Nordbø, Kristine Karlsrud Berg, Randi Eikeland.

**Supervision:** Yngvar Tveten, Hege Kersten, Anne Kristin Møller Fell.

**Validation:** Marjut Sarjomaa, Lien My Diep, Chi Zhang, Yngvar Tveten, Harald Reiso, Carina Thilesen, Svein Arne Nordbø, Kristine Karlsrud Berg, Ingeborg Aaberge, Neil Pearce, Hege Kersten, Jan Paul Vandenbroucke, Randi Eikeland, Anne Kristin Møller Fell.

**Visualization:** Marjut Sarjomaa, Randi Eikeland, Anne Kristin Møller Fell.

**Writing – original draft:** Marjut Sarjomaa, Anne Kristin Møller Fell.

**Writing – review & editing:** Marjut Sarjomaa, Lien My Diep, Chi Zhang, Yngvar Tveten, Harald Reiso, Carina Thilesen, Svein Arne Nordbø, Kristine Karlsrud Berg, Ingeborg Aaberge, Neil Pearce, Hege Kersten, Jan Paul Vandenbroucke, Randi Eikeland, Anne Kristin Møller Fell.

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
