## [Decision Letter · Decision Letter 0]

31 May 2022

PONE-D-22-04506SARS-CoV-2 antibody persistence after five and twelve months: A cohort study from South-Eastern NorwayPLOS ONE

Dear Dr. Sarjomaa,

Thank you for submitting your manuscript to PLOS ONE. After careful consideration, we feel that it has merit but does not fully meet PLOS ONE’s publication criteria as it currently stands. Therefore, we invite you to submit a revised version of the manuscript that addresses the points raised during the review process. There are few comments that raised by the reviewers. The authors are required to response to these comments. 

We look forward to receiving your revised manuscript.

Kind regards,

Mohammed Abdelfatah Mosa Alhoot, PhD

Academic Editor

PLOS ONE

Journal Requirements:

Reviewers' comments:

Reviewer's Responses to Questions

**Comments to the Author**

1. Is the manuscript technically sound, and do the data support the conclusions?

Reviewer #1: Yes

Reviewer #2: No

Reviewer #3: Yes

2. Has the statistical analysis been performed appropriately and rigorously? 

Reviewer #1: Yes

Reviewer #2: Yes

Reviewer #3: Yes

3. Have the authors made all data underlying the findings in their manuscript fully available?

Reviewer #1: No

Reviewer #2: Yes

Reviewer #3: No

4. Is the manuscript presented in an intelligible fashion and written in standard English?

Reviewer #1: Yes

Reviewer #2: Yes

Reviewer #3: Yes

5. Review Comments to the Author

Reviewer #1: Data need to be available fully as data policy of plos one, description of other data need to be describe fully. Manuscript need to address the utility of the 3 times PCR detection in individuals need to specified.

Reviewer #2: This study has only measured antibodies and even if they do mention their limitations, I find this not enough to make it clear that in case of COVID-19 there has been found no clear correlation between antibodies and protection.

Moreover, the recall bias could be really very high and skewed to the very mild cases and conclusion could be prone to misinterpretation against the need of high levels of vaccine coverage, making it technically not sound or relevant.

Reviewer #3: 1/ what was the purpose of including PCR -ve participants ?

2/ Inclusion criteria not so clear as authors mentioned patients with symptoms and this can be wide from mild to severe cases

3/ study design not clear and confusing to have matched cohort study , need more clarification on why choosing this design

6. PLOS authors have the option to publish the peer review history of their article (what does this mean?). If published, this will include your full peer review and any attached files.

Reviewer #1: No

Reviewer #2: No

Reviewer #3: No

---

## [Author Response · Author response to Decision Letter 0]

6 Jul 2022

Author response

July 6, 2022

PONE-D-22-04506 SARS-CoV-2 antibody persistence after five and twelve months: A cohort study

from South-Eastern Norway

Dear editor,

Thank you for the invitation to submit a revised version of the manuscript “SARS-CoV-2 antibody persistence after five and twelve months: A cohort study from South-Eastern Norway” to PLOS ONE.

We greatly appreciated the constructive feedback we received from the reviewers. We have responded to each point in the reviews.

Editorial comments

RESPONSE: We have revised the style requirements and file naming. We have revised the tables to PLOS ONE style, removed the empty rows from the tables and quality checked the numbers. We made some minimal changes in decimals. All changes are shown in “manuscript with track changes”. We have revised the caption to the table 4 to describe it better. We noticed that two rows for the reference group for gender and BMI had fallen out and are now replaced in the table 4.

2. In your Data Availability statement, you have not specified where the minimal data set underlying the results described in your manuscript can be found. PLOS defines a study's minimal data set as the underlying data used to reach the conclusions drawn in the manuscript and any additional data required to replicate the reported study findings in their entirety. All PLOS journals require that the minimal data set be made fully available.

RESPONSE: We have now written a new Data Availability Statement:” There are legal and ethical restrictions on sharing our dataset. The project is approved by the Regional Committees for Medical and Health Research Ethics (ID 146469), and by the Data Protection Officers in the participating Hospitals. Our data set is not fully anonymized and has a relative small sample size making identification of individuals possible. The potentially identifying patient information is age, birthdate, location and dates for PCR and antibody tests. However, data requests for the minimal dataset, which includes only the main variables of the final analyses, can be made to the Research Department at The Telemark Hospital Trust, Ulefossvegen 55, 3710 Skien, Norway email: fou@sthf.no.

3. Please review your reference list to ensure that it is complete and correct. 

RESPONSE: Thank you-we have updated reference 23 and 30 from a preprint to a published version. There are no retracted papers in the reference list.

Reviewer #1: Data need to be available fully as data policy of plos one, description of other data need to be describe fully. Manuscript need to address the utility of the 3 times PCR detection in individuals need to specified.

Author: Answer to the first comment: Please see our response to the comment 2 above. Thank you for the last comment; we have now clarified this point. We included participants after they were tested with PCR and only used the first PCR test they performed in the inclusion period. Hence, only one PCR-test was used, and thereafter we collected blood samples for antibody testing. We have now included a statement about this in the manuscript on pages 3-4; lines 113-116: “The study included SARS-CoV-2 RT-PCR -positive and -negative participants regardless of symptoms. We included participants who performed a PCR test in the inclusion period, and used the first PCR test result”.

Reviewer #2: This study has only measured antibodies and even if they do mention their limitations, I find this not enough to make it clear that in case of COVID-19 there has been found no clear correlation between antibodies and protection.

Moreover, the recall bias could be really very high and skewed to the very mild cases and conclusion could be prone to misinterpretation against the need of high levels of vaccine coverage, making it technically not sound or relevant.

Author: Thank you for your important comments. In our study, we used self-reported questionnaires to assess reinfection. None of the 391 SARS-CoV-2 PCR positive participants reported reinfections until 10-12 months follow-up. We asked questions about SARS-CoV-2 reinfection and symptoms from the airways 3-5 months and 10-12 months after the first positive PCR. 

The study was conducted early in the pandemic, and included the first and second pandemic wave, which was dominated by Alfa-SARS-CoV-2 virus in Norway. This virus variant was less infectious, but more pathogenic than the current Omicron variant. We have studied only the total antibody levels against spike protein in non-vaccinated participants, and it was not possibly to study protective immunity after COVID-19 and vaccination in this part of our study. We have adjusted the discussion and the conclusion to meet this comment on page 12-13, lines 342-343 and 355-357: “Measurements of cellular immunity and neutralising antibodies were not feasible in this study, hence no firm conclusions can be drawn regarding the immune status post COVID-19. Our results indicate that the level of total antibodies against spike protein following mild COVID-19 persists for at least 10 to 12 months”. 

Recall bias is possible in our study and is a general problem with questionnaire-based studies, but early in the pandemic in Norway people were tested frequently with PCR if they had new symptoms and PCR tests were available. We included a question regarding PCR retesting in our questionnaire. To find out if the participants had an asymptomatic reinfection it is now possible to use nucleocapsid antibody test, but this was unfortunately not available in the first part of our study. 

None of the 391 PCR positive and 791 PCR negative participates were vaccinated before the first PCR test because the official vaccination in Norway started first 26.12.2020. and the few participants, who were vaccinated before their antibody tests, were excluded from the study.

Reviewer #3: 

1/ What was the purpose of including PCR -ve participants?

Author: Thank you for the important question. Our study is the first part of the COVITA -project (COVID-19 Telemark and Agder study). In the current study, we included PCR- participants to present population characteristics and the presence of antibodies in this group, as well as the “background” rates of infection in the population. We found that there were few PCR-negative participants with antibodies, which may confirm the high specificity of the PCR tests. We have adjusted the text to respond to this comment on page 3, lines 108-110 and 116-119: “We here present population characteristics and antibody persistence of PCR -positive (PCR+) and PCR -negative (PCR-) participants. Further, results from the first follow-up of the PCR+ participants regarding possible predictors for antibody change are shown. For each PCR+ participant, we aimed to select two PCR- participants matched by residency and time for the PCR test to reduce the probability of health care-seeking bias. In this study, we included PCR- participants to allow comparison of population characteristics between PCR+ and PCR - participants, and to assess presence of antibodies also among PCR- participants”.

2/ Inclusion criteria not so clear as authors mentioned patients with symptoms and this can be wide from mild to severe cases

Author: We have now adjusted the text on page 3, lines 111-12 and 113-116: “Adults aged 18 years or older residing in South-Eastern Norway (Agder and Telemark counties) were considered eligible for inclusion in the study. The study included SARS-CoV-2 RT-PCR -positive and -negative participants regardless of symptoms. We included participants who performed a PCR test in the inclusion period, and used the first PCR test result”. Most of the participants had mild symptoms reflecting the pandemic in Norway. 

3/ Study design not clear and confusing to have matched cohort study, need more clarification on why choosing this design

Author: Thank you for this comment. We have now adjusted the text on pages 3-4, lines 107-110 and 116-117. “This study is the first part of the COVITA -project (COVID-19 Telemark and Agder study) which is a prospective multi-centre cohort study. We here present population characteristics and antibody persistence of PCR -positive (PCR+) and PCR -negative (PCR-) participants. Further, results from the first follow-up of the PCR+ participants regarding possible predictors for antibody change are shown. For each PCR+ participant, we aimed to select two PCR- participants matched by residency and time for the PCR test to reduce the probability of health care-seeking bias” (1).

1. Pearce N, Vandenbroucke JP, VanderWeele TJ, Greenland S. Accurate Statistics on COVID-19 Are Essential for Policy Guidance and Decisions. American journal of public health. 2020:e1-e3.

Matching in a cohort design tries to achieve the same that is aimed for by randomization: making the groups that will be compared as much alike as possible. More generally, matching in cohort studies is a very common design, which is currently being used in many different COVID-19 studies, and in fact is regarded as the “gold standard” approach for observational studies. 

2. Dickerman BA, Gerlovin H, Madenci Al, Kurgansky KE, Ferolito BR, Figueroa Muniz MJ et al. Comparative Effectiveness of BNT162b2 and mRNA-1273 Vaccines in U.S. Veterans. The New England journal of medicine. 2022;386 (2):105- 15.https://www.nejm.org/doi/full/10.1056/NEJMoa2115463

Thank you very much for the feedback we received. We think that the manuscript is significantly improved after revision and hope that our responses are satisfactory. We look forward to receiving your reply.

Yours sincerely,

Marjut Sarjomaa

MD

Telemark Hospital Trust

Skien, Norway

---

## [Decision Letter · Decision Letter 1]

27 Jul 2022

SARS-CoV-2 antibody persistence after five and twelve months: A cohort study from South-Eastern Norway

PONE-D-22-04506R1

Dear Dr. Sarjomaa,

We’re pleased to inform you that your manuscript has been judged scientifically suitable for publication and will be formally accepted for publication once it meets all outstanding technical requirements.

Kind regards,

Mohammed Abdelfatah Mosa Alhoot, PhD

Academic Editor

PLOS ONE

Additional Editor Comments (optional):

Reviewers' comments:

Reviewer's Responses to Questions

**Comments to the Author**

1. If the authors have adequately addressed your comments raised in a previous round of review and you feel that this manuscript is now acceptable for publication, you may indicate that here to bypass the “Comments to the Author” section, enter your conflict of interest statement in the “Confidential to Editor” section, and submit your "Accept" recommendation.

Reviewer #3: All comments have been addressed

2. Is the manuscript technically sound, and do the data support the conclusions?

Reviewer #3: Yes

3. Has the statistical analysis been performed appropriately and rigorously? 

Reviewer #3: Yes

4. Have the authors made all data underlying the findings in their manuscript fully available?

Reviewer #3: Yes

5. Is the manuscript presented in an intelligible fashion and written in standard English?

Reviewer #3: Yes

6. Review Comments to the Author

Reviewer #3: (No Response)

7. PLOS authors have the option to publish the peer review history of their article (what does this mean?). If published, this will include your full peer review and any attached files.

Reviewer #3: No

---

## [Editor Report · Acceptance letter]

1 Aug 2022

PONE-D-22-04506R1 

SARS-CoV-2 antibody persistence after five and twelve    months: A cohort study from South-Eastern Norway 

Dear Dr. Sarjomaa:

I'm pleased to inform you that your manuscript has been deemed suitable for publication in PLOS ONE. Congratulations! Your manuscript is now with our production department. 

Kind regards, 

on behalf of

Dr. Mohammed Abdelfatah Mosa Alhoot 

Academic Editor

PLOS ONE